# Combined Midportion Achilles and Plantaris Tendinopathy: A 1-Year Follow-Up Study after Ultrasound and Color-Doppler-Guided WALANT Surgery in a Private Setting in Southern Sweden

**DOI:** 10.3390/medicina59030438

**Published:** 2023-02-23

**Authors:** Håkan Alfredson, Markus Waldén, David Roberts, Christoph Spang

**Affiliations:** 1Department of Community Medicine and Rehabilitation, Sports Medicine, Umeå University, 90187 Umeå, Sweden; 2Alfredson Tendon Clinic, Capio Ortho Center Skåne, 21532 Malmö, Sweden; 3Capio Ortho Center Skåne, 21532 Malmö, Sweden; 4Department of Health, Medicine and Caring Sciences, Linköping University, 58183 Linköping, Sweden; 5Department of Sports Science, University of Würzburg, 97074 Sanderring, Germany; 6Integrative Medical Biology, Anatomy Section, Umeå University, 90187 Umeå, Sweden

**Keywords:** Achilles tendinopathy, plantaris tendinopathy, surgical treatment, follow-up

## Abstract

*Background and Objectives*: Chronic painful midportion Achilles combined with plantaris tendinopathy can be a troublesome condition to treat. The objective was to prospectively follow patients subjected to ultrasound (US)- and color doppler (CD)-guided wide awake, local anesthetic, no-tourniquet (WALANT) surgery in a private setting. *Material and Methods*: Twenty-six Swedish patients (17 men and 9 women, mean age 50 years (range 29–62)) and eight international male patients (mean age of 38 years (range 25–71)) with combined midportion Achilles and plantaris tendinopathy in 45 tendons altogether were included. All patients had had >6 months of pain and had tried non-surgical treatment with eccentric training, without effect. US + CD-guided surgical scraping of the ventral Achilles tendon and plantaris removal under local anesthesia was performed on all patients. A 4–6-week rehabilitation protocol with an immediate full-weight-bearing tendon loading regime was used. The VISA-A score and a study-specific questionnaire evaluating physical activity level and subjective satisfaction with the treatment were used for evaluation. *Results:* At the 1-year follow-up, 32/34 patients (43 tendons) were satisfied with the treatment result and had returned to their pre-injury Achilles tendon loading activity. There were two dropouts (two tendons). For the Swedish patients, the mean VISA-A score increased from 34 (0–64) before surgery to 93 (61–100) after surgery (*p* < 0.001). There were two complications, one wound rupture and one superficial skin infection. *Conclusions:* For patients suffering from painful midportion Achilles tendinopathy and plantaris tendinopathy, US + CD-guided surgical Achilles tendon scraping and plantaris tendon removal showed a high satisfaction rate and good functional results 1 year after surgery.

## 1. Introduction

Midportion Achilles tendinopathy is relatively common in the general population [1,2,3,4] and is known to be a problematic injury for runners [5,6,7]. The etiology is unknown, but overuse among active and non-actives is generally considered to be the main causative factor [5,6,7]. The condition has also been shown to be relatively common among individuals suffering from diagnoses involved in the metabolic syndrome with high blood lipids, type 2 diabetes, and hypertension [8]. Men and women are equally affected [1].

Treatment is known to be difficult, but painful eccentric calf muscle training has been shown to be an efficient non-surgical method [9,10,11,12]. Multiple different surgical methods have been described, but most of them are tendon invasive and require long rehabilitation [13,14,15,16,17,18,19,20]. Studies on surgical biopsies have shown nerves in close relation to blood vessels outside but not inside the tendons [21,22]. These findings have been used for the development of the ultrasound (US)- and color doppler (CD)-guided surgical scraping method outside the ventral side of the Achilles tendon [23,24]. More recently, the plantaris tendon has shown similar tendinopathic changes as in the Achilles tendon, but also sensory nerves inside and outside the plantaris tendon [25,26]. The plantaris tendon can also mechanically interfere with the Achilles tendon [27]. For patients with insufficient symptom relief from non-surgical treatment methods, US + CD-guided combined Achilles scraping targeting the regions with high blood flow and nerves outside the tendon and local plantaris tendon removal has been shown to be successful in many patients [28,29,30].

Midportion Achilles tendinopathy is a debilitating condition among individuals on multiple different activity levels, and there is an obvious need to find an appropriate treatment method. Avoiding intratendinous surgery has major advantages, and new research has opened the possibility of using methods focusing on the outside of the Achilles. The aim of this prospective study was to introduce the US + CD-guided wide awake, local anesthetic, no-tourniquet (WALANT) surgery for patients suffering from chronic painful midportion Achilles combined with plantaris tendinopathy in a private setting in southern Sweden and evaluate the 1-year functional outcome and subjective patient satisfaction with the treatment.

## 2. Materials and Methods

### 2.1. Patients and Clinical Examination

Informed written consent was obtained from all patients. Twenty-six consecutive Swedish patients (17 men and 9 women, mean age 50 years (range 29–62), were included (Table 1) between August 2020 and March 2022. The patients’ activity levels were recreational athletes (*n* = 12); non-actives (*n* = 5); padel (*n* = 4); gym (*n* = 1); and elite athletes participating in soccer (*n* = 2), triathlon (*n* = 1), and orienteering (*n* = 1). In a separate group during the same study period, 8 international male patients (11 tendons) were also included and operated on. There were 6 professional athletes involved in soccer (*n* = 4), cross-country skiing (*n* = 1), and hurling (*n* = 1), as well as 2 high-level recreational athletes involved in marathon running (*n* = 1) and dancing (*n* = 1). There were no smokers among the patients, and 4 patients had treatment for hypertension.

Inclusion criteria: adult patients (≥18 years) with long duration (>6 months) of midportion Achilles tendon pain and eccentric training without effect, or inability to perform eccentric training because of back, hip, knee, ankle, or foot problems. Exclusion criteria: chronic systemic inflammatory conditions and previous surgery inside or close to the Achilles tendon.

Patients were examined and operated on at Capio Orthocenter Skåne in Malmö, Sweden. Clinical examination demonstrated a thickening of the Achilles midportion with tenderness located on both the ventral and the medial side of the tendon thickening in all cases. High resolution gray-scale US + CD (S-500, Siemens AG, Germany) using a linear multifrequency (8–13 MHz) probe showed a thickened Achilles midportion (>7 mm) with irregular tendon structure. On the medial side of the Achilles midportion, there was a wide or thick plantaris tendon (Figure 1). There was also a localized high blood flow outside and inside the regions with structural tendon changes.

### 2.2. Midportion Achilles Tendon Scraping and Local Plantaris Tendon Removal

Same-day pre-operative US + CD examination was carried out by the first author for verification and applying skin markers to map the **region** with maximum palpation tenderness, the localized high blood flow outside the deep side of the tendon, and where the plantaris tendon was positioned close to the medial side of the Achilles.

After disinfecting the skin with wet cloths of chlorhexidine cutaneous solution (Klorhexidionsprit 5 mg/mL, Fresenius Kabi, Germany), 4–7 mL of a local anesthetic (Xylocain + adrenalin 10 mg/mL + 5 μg/mL, Aspen, South Africa) was injected on the medial and ventral side (the mapped region) of the Achilles midportion. The skin was then scrubbed and draped with a sterile paper-cover exposing only the midportion of the Achilles tendon.

A longitudinal skin incision (1–2 cm) was placed on the medial side of the Achilles midportion. Following blunt dissection, the plantaris tendon was carefully identified (Figure 2), released, and followed distally and proximally from the skin incision. The plantaris was cut in both ends, resulting in 5–8 cm of the tendon being extirpated. Any vascularized fat tissue inter-positioned between the Achilles and the plantaris tendons was scraped. Then, the traditional Achilles scraping procedure was performed [23,24,28]. In the regions with US + CD-verified high blood flow outside the ventral and medial side of the tendon, the tendon was completely released from the ventral soft tissue. This scraping procedure was performed by sharp dissection using a scalpel, staying close to the ventral tendon. Following careful hemostasis using bi-polar diatermia, the skin was closed by single non-resorbable sutures, which were removed after 3 weeks.

### 2.3. Postoperative Rehabilitation

There was clinical and US + CD follow-up 3 and 6 weeks after the operation. Then, extra follow-up only if there were complications. 

Day **1**: Surgery day: rest, elevated foot;Day 2: ROM (range of movement) exercises and short, full-weight-bearing walks;Day 3–7: Gradually increased walking activity, light seated stretching (bent knee);Day 8–21: Start light bicycling; longer and faster walks;Week 4: Free walking and biking;Week 5: Introduce jogging: walk 50 m, jog 50 m; walk 50 m, jog 100 m; etc.;Week 6: Jogging–running.

The described protocol was based on previous studies [23,24,28,29].

### 2.4. Outcome Measures

The self-administered Victorian Institute of Sports Assessment Achilles (VISA-A) functional score [30] and a study-specific questionnaire evaluating subjective satisfaction with treatment (satisfied or not satisfied), physical activity level, medication, sick leave, and complications were used for evaluation. The VISA-A score was completed by the patients on the day of surgery (pre-treatment) and after 1 year (post-treatment). The study-specific questionnaire was completed only at the 1-year follow-up. In the sub-group of international athletes, follow-up was restricted to mail contact evaluating subjective satisfaction with the treatment and return to their sport activity. It was not possible to get VISA-A scoring.

### 2.5. Ethical Consideration

Ethics approval was received from the Ethical Board in Uppsala-dnr 2022-02889-01. Informed written consent was obtained from all patients. 

### 2.6. Statistical Methods

SPSS (Statistical Package of Social Science) that has been shown to be useful and reliable [30] was used to analyze the data (SPSS Inc., Chicago, IL, USA). All calculations were measured at the group level. Paired Student’s t-test was selected to identify differences before and after surgery. The level of significance was set to *p* < 0.05.

## 3. Results

At the 1-year follow-up, 32/34 patients (43 tendons) were satisfied with the treatment. There was a significant (*p* < 0.001) increase in the mean VISA-A score from 34 (range 0–64) before surgery to 93 (range 61–100) after surgery (see also Table 2).

There were two dropouts (two tendons). These patients did not answer the letter, telephone calls, or text messages. From the media, we know that one of these patients played professional soccer 2 months after the operation. 

In the separate group with international athletes, all were satisfied and back in full activity in their sport.

There was one wound rupture that needed a re-operation, and one superficial skin infection successfully treated with antibiotics.

## 4. Discussion

This 1-year follow-up study on 34 patients suffering from combined midportion Achilles and plantaris tendinopathy in 45 tendons and undergoing WALANT surgery with US + CD-guided Achilles tendon scraping and local plantaris removal showed high satisfaction rates and good functional results.

### 4.1. Patients and Main Findings

The patients in the current study represent two different groups. The larger group containing 26 patients is from the general population in the most southern part of Sweden, referred from various insurance companies. The smaller group consisted of eight male international athletes referred directly from club/federation medical staff or via the athletes themselves. Altogether, it is a generally healthy group of patients who all had a long duration of tendon pain not responding to non-surgical treatment including eccentric training. The activity levels varied from mainly recreational actives in the Swedish group to elite athletes in the international group. The US + CD-guided surgical scraping procedure with local plantaris removal has in previous studies in northern Sweden been shown to be a successful treatment method [23,24,28,29,30]. In the current study, similar subjective satisfaction and good functional results were reproduced in a private setting involving patients referred from insurance companies in the very south of Sweden.

### 4.2. Plantaris Involvement

Treatment with eccentric training is generally efficient for patients suffering from chronic painful midportion Achilles tendinopathy [9,10,11,12]. In a minor group of patients, symptoms do not improve with gradual tendon loading regimes, and a possible reason to lack of effect of eccentric training could be mechanical interference from the plantaris tendon [26,27,31,32]. Our experiences over the years with the treatment of patients with midportion Achilles tendinopathy are that if the plantaris tendon is involved, there is often a paradoxical worsening with eccentric training with localized sharp pain on the medial side of the Achilles tendon.

Plantaris tendinopathy together with Achilles tendinopathy has been demonstrated in previous studies [25,26,27,31,33]. The plantaris tendon was found to be thickened, located close to, and seemingly interfering with the medial side of the Achilles tendon [28,31,34]. The findings from these initial studies led to the development of a surgical method where removal of the plantaris was performed routinely in cases where it was located close to the medial Achilles at the site of the symptoms. Histological examination of the excised thickened plantaris tendons showed similar tendinosis changes, as has been identified in Achilles tendinopathy [25,26,35]. In addition, the richly vascularized fat tissue between the Achilles and the plantaris tendons was found to be richly innervated, often by multiple sensory nerves [35].

### 4.3. Surgical Procedures

Traditional surgical treatment methods for midportion Achilles tendinopathy include different approaches for debridement with excision of tendinopathic regions inside the tendon [13,14,15,16,17], sometimes combined with flexor hallucis longus transfer procedures and multiple longitudinal splitting [18,19,20]. All these procedures are tendon invasive and require periods with low load or even immobilization after surgery with accompanying long rehabilitation periods before return to full Achilles-tendon-loading activities. The main advantage of the US + CD-guided surgical procedure used in this study is that the operation is performed outside the tendon, not directly affecting the Achilles tendon structure, allowing for immediate full weight bearing loading. This opens up the possibility for an accelerated rehabilitation and also minimizing the potential risks related to immobilization. Most patients in the current study were back in full Achilles tendon loading activities within 6–8 weeks, and only one patient (with the wound rupture) needed to be on sick leave after the operation. Another advantage is that the surgery is performed under local anesthesia (WALANT), thereby also avoiding risks related to general or spinal anesthesia. Finally, US + CD guidance localizes the exact target for the surgical procedure, making it possible to minimize the skin incision and the tissue trauma during the procedure.

### 4.4. Complications

There were two complications among the 45 operated Achilles and plantaris tendons. The most serious complication was a wound rupture occurring one week after suture removal (4 weeks after surgery). There was no infection involved and no direct trauma or heavy Achilles tendon loading, but for some unknown reason, there was slow skin healing. After healing, another **minor** operation in local anesthetics was needed to remove excessive scar tissue affecting the Achilles tendon movement. Wound healing was uncomplicated following this subsequent surgery, and the patient was back in full training after six months. Although there was only one wound rupture in this sample, the first author’s experience from operating on a large number of individuals with this diagnosis in other settings is that delayed skin healing can sometimes occur. As a precaution, we therefore nowadays do not remove the sutures before three weeks and carefully inform the patients about proper wound care. In another patient, there was a superficial skin infection (staphylococcus) successfully treated with antibiotics without surgical debridement needed. Two patients had a prolonged rehabilitation period, and heavy Achilles tendon loading could not be started until 6 months after the operation. In these two patients, who were carefully followed with repeated US + CD examinations, there were no partial ruptures visible, but the Achilles tendon was swollen with a high blood flow in the whole tendon, and the Achilles tendon was sensitive to load. This is another experience from the first author that there can be a delayed healing response for unknown reasons before the patients are pain-free and can return to full Achilles tendon loading activities. Therefore, follow-up after surgery is essential. In a study by Maffulli et al., it was shown that the results after surgery were worse in a non-athletic population [20]. In the current study, there were no differences in the results between non-actives, recreationally actives, and elite level actives.

### 4.5. Limitations

A weakness in the current study is that it was only a 1-year follow up. The results could maybe have changed over time, but our experiences from other studies on this surgical method **are** that the failures tend to show up within the first six months, and longer-term follow-up studies have shown stable good results [28,29]. Another weakness is the use of only a questionnaire follow-up. It would have been ideal to also perform a clinical and ultrasound follow-up, but this was unfortunately not possible due to logistical reasons. It can be discussed as to whether randomized studies comparing the results after the US + CD method with other surgical methods are needed, but we believe it would have been an ethical dilemma to use tendon-invasive operations when operation outside the tendon in multiple studies have shown very high success rates [23,24,28,29].

## 5. Conclusions

For treatment of midportion Achilles tendinopathy, avoiding intratendinous surgery has major advantages, and new research has opened up the possibility of using surgical treatment methods targeting the outside of the Achilles. On the basis of research findings on innervation patterns for chronic painful midportion Achilles tendinopathy and plantaris tendon involvement, this 1-year follow-up study on insurance patients in the south of Sweden and international elite athletes showed good functional effects and allowed for a quick return to Achilles tendon loading activities after treatment with US + CD-guided surgical scraping outside the tendon combined with local plantaris tendon removal. 

## Figures and Tables

**Figure 1 medicina-59-00438-f001:**
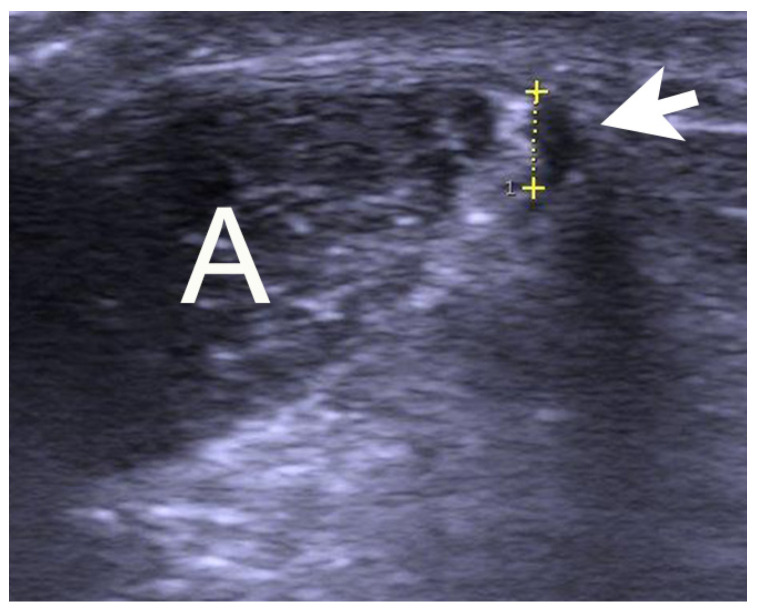
Gray-scale ultrasound picture of a patient suffering from midportion Achilles and plantaris tendinopathy showing a widened plantaris tendon (arrow/yellow scale) located close to the medial side of the Achilles tendon A.

**Figure 2 medicina-59-00438-f002:**
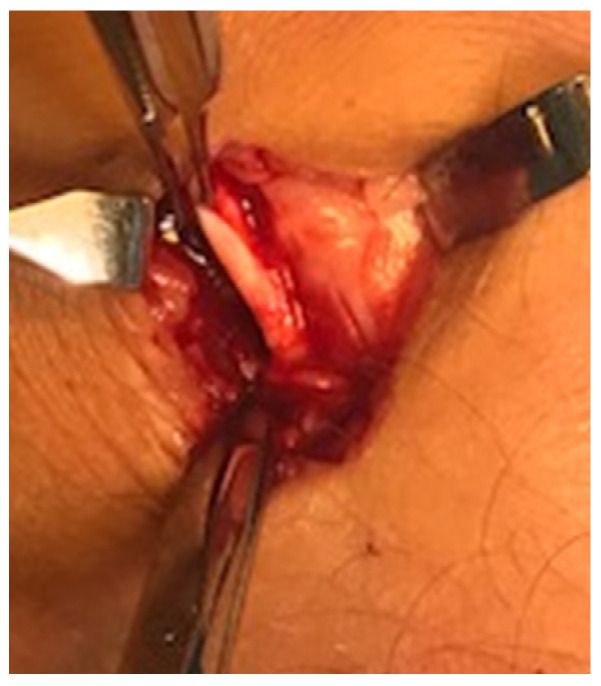
Surgery of a patient with midportion Achilles and plantaris tendinopathy. There is a thick and wide plantaris tendon located close to the medial side of a thickened Achilles midportion.

**Table 1 medicina-59-00438-t001:** Patients’ characteristics.

Values	Females	Males	Total
Age	51 (29–61)	49 (32–62)	50 (29–62)
Height	168 (164–173)	185 (179–191)	181 (164–191)
Weight	65 (58–78)	88 (80–104)	80 (58–104)

**Table 2 medicina-59-00438-t002:** Comparison of VISA-A baseline measure and follow-up comparison (mean (range)).

Baseline	Follow-Up	*p*
34 (0–64)	93 (61–100)	<0.001

## Data Availability

The data presented in this study are available on request from the corresponding author.

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
