# Peer review of "Combined Midportion Achilles and Plantaris Tendinopathy: A 1-Year Follow-Up Study after Ultrasound and Color-Doppler-Guided WALANT Surgery in a Private Setting in Southern Sweden"

_medicina, 2023, doi:10.3390/medicina59030438_

Round 1

Reviewer 1 Report

Thanks for giving me the opportunity to review your paper. It deals with an exciting and relevant topic.

In the introduction expanded reflection on the need of the study and topic background is needed.

In methodology, the references for the rehabilitation protocol and reliability of the scale used should be added.

The result section should add a table for patients’ characteristics, outcome measure baseline, and follow-up comparison.

You should update the discussion with recent literature and revise the conclusion to enhance the significance of the study.

I suggest citing and referencing recent articles.

Reviewer 2 Report

Dear Authors,

Your study provides valuable insights into the long-term outcomes of ultrasound and color doppler-guided WALANT surgery for midportion Achilles and plantaris tendinopathy. The 12-month follow-up period gives us a good understanding of the effectiveness of this surgical intervention.

I also appreciate the level of detail you provide on the methodology used in the study, which enhances the scientific rigor and reliability of your findings. The clear presentation of results and thorough statistical analysis also make your paper a valuable resource for other researchers and clinicians interested in sports medicine.
